# Effect of PM2.5 Levels on ED Visits for Respiratory Causes in a Greek Semi-Urban Area

**DOI:** 10.3390/jpm12111849

**Published:** 2022-11-05

**Authors:** Maria Mermiri, Georgios Mavrovounis, Nikolaos Kanellopoulos, Konstantina Papageorgiou, Michalis Spanos, Georgios Kalantzis, Georgios Saharidis, Konstantinos Gourgoulianis, Ioannis Pantazopoulos

**Affiliations:** 1Department of Emergency Medicine, Faculty of Medicine, University of Thessaly, BIOPOLIS, 41110 Larissa, Greece; 2Department of Anesthesiology, Faculty of Medicine, University of Thessaly, BIOPOLIS, 41110 Larissa, Greece; 3Department of Respiratory Medicine, Faculty of Medicine, University of Thessaly, BIOPOLIS, 41110 Larissa, Greece; 4Department of Mechanical Engineering, University of Thessaly, Leoforos Athinon, 8 Pedion Areos, 38334 Volos, Greece

**Keywords:** PM2.5, air pollution, respiratory diseases, asthma, pneumonia, upper respiratory infections, emergency department

## Abstract

Fine particulate matter that have a diameter of <2.5 μm (PM2.5) are an important factor of anthropogenic pollution since they are associated with the development of acute respiratory illnesses. The aim of this prospective study is to examine the correlation between PM2.5 levels in the semi-urban city of Volos and Emergency Department (ED) visits for respiratory causes. ED visits from patients with asthma, pneumonia and upper respiratory infection (URI) were recorded during a one-year period. The 24 h PM2.5 pollution data were collected in a prospective manner by using twelve fully automated air quality monitoring stations. PM2.5 levels exceeded the daily limit during 48.6% of the study period, with the mean PM2.5 concentration being 30.03 ± 17.47 μg/m^3^. PM2.5 levels were significantly higher during winter. When PM2.5 levels were beyond the daily limit, there was a statistically significant increase in respiratory-related ED visits (1.77 vs. 2.22 visits per day; *p*: 0.018). PM2.5 levels were also statistically significantly related to the number of URI-related ED visits (0.71 vs. 0.99 visits/day; *p* = 0.01). The temperature was negatively correlated with ED visits (r: −0.21; *p* < 0.001) and age was found to be positively correlated with ED visits (r: 0.69; *p* < 0.001), while no statistically significant correlation was found concerning humidity (r: 0.03; *p* = 0.58). In conclusion, PM2.5 levels had a significant effect on ED visits for respiratory causes in the city of Volos.

## 1. Introduction

Anthropogenic air pollution is a major cause of environmental pollution, which can have detrimental effects on human health [1]. Fine particle matter 2.5 (PM2.5) are fine particles that have an aerodynamic diameter of less than 2.5 μm, which are produced mainly by wood and fuel combustion [2]. Because of their small diameters, they are able to infiltrate through the nose into the lower respiratory system, accumulating in respiratory bronchioles [2]. Alveolar damage may consequently occur due to the production of free radicals, imbalanced intracellular homeostasis and inflammation [2].

High levels of PM2.5 have been previously associated with several respiratory diseases, as well as increased morbidity and mortality due to respiratory causes [3]. In a meta-analysis of 16 studies, short-term exposure to increased PM2.5 levels were shown to precipitate asthma attacks leading to increased hospitalizations in patients suffering from severe respiratory disorders [4]. Moreover, a prospective study of 431 patients suffering from Chronic Obstructive Pulmonary Disease (COPD) in a highly polluted city of Italy showcased that high PM10 and PM2.5 levels may precipitate COPD exacerbations and increase hospital admissions in these patients. Furthermore, several prospective studies and a meta-analysis by Kim et al. proved that elevated PM2.5 levels have been associated with a surge in Emergency Department (ED) visits for respiratory causes, such as asthma attacks and respiratory infections [4,5,6].

While the connection between PM2.5 pollution and respiratory diseases has been extensively researched, there is a lack of studies pertaining to the relationship between air pollution and health in rural or semi-urban areas. Although these areas may have lower exposure to industrial and motor vehicle pollution [7], studies have shown that several non-urban regions are greatly exposed to pollution produced by agriculture, coal burning and mining and wood burning [8,9]. The semi-urban city of Volos is a medium-size coastal city in Greece, where high levels of PM10 and PM2.5 pollution have been previously recorded [10,11]. Moustris et al. studied the effect of PM10 pollution in hospital admissions for respiratory diseases during a five-year period in Volos [10]. Their results showcased that increased PM10 levels were associated with an 25% increase in hospital admissions. Later, our team in a similarly designed study studied the effect of PM2.5 levels in pediatric ED visits for respiratory causes during a one-year period [11]. Likewise, elevated PM2.5 levels resulted in an increase in pediatric ED visits for respiratory causes. 

The aim of the present study was to investigate the relationship between daily PM2.5 levels in the semi-urban Greek city of Volos and the number of adult ED visits for respiratory causes. 

## 2. Materials and Methods

### 2.1. Study Design and Population

This study retrospectively analysed a prospectively collected database. The study was approved by the local scientific committee (2/7-2-2019). The methodology has been previously described in detail for our pediatric ED visits study [11]. 

Data were collected for adult patients visiting the ED of Volos with respiratory symptoms, including asthma, pneumonia, COPD and upper respiratory tract infection (URI)-related visits between 1 March 2019 and 29 February 2020. We used the following inclusion criteria: (1) residents of Volos city, (2) those who visited the adult ED and (3) those with respiratory conditions (URI, COPD exacerbation, asthma exacerbation and pneumonia). The following data were collected for all patients: age, gender, address of residence, main complaint/diagnosis and date of visit. 

### 2.2. Network of Sensors

The GreenYourAir research group established a fully automated network for monitoring air pollution in the city of Volos. Twelve stations located throughout the central and greater Volos areas collected the data prospectively during the entire study period. The locations were chosen based on a mathematical formula and an optimization model developed by the GreenYourAir team that parcellated the city into smaller areas. Briefly, the team divided the city into five distinct areas, namely the commercial and recreational zone; the industrial zone; and the high-density, medium-density and low-density residential zones, while the traffic jam of the city was divided into three categories (high, medium and low). Furthermore, additional information regarding the city was taken into consideration, such as the geographical and geomorphological characteristics of the area, the commercial port, the passenger port, the urban and intercity bus stations, the train station, the main roads, the recreational parks, the sport facilities, the schools and other academic institutions and two big industrial plants located just outside the city (one cement company and one petroleum company). It should be mentioned that the residents of the city use oil, natural gas and fireplaces for heating during the winter. The exact location of the sensors throughout the city of Volos is presented in Figure 1. 

The devices that were used for monitoring (GreenYourAir Device 1178/PM2.5) used light-scattering methods for data collection, as described previously in the literature [12,13]. The devices consist of a sensor, an expansion shield and an Arduino YUN rev. 2. The sensor collects data regarding PM2.5 concentration, temperature and humidity in the area. The collection of data was performed automatically every second for the entire 24 h of the day. The programming language used for the devices was C++.

A two-phase calibration methodology was implemented to check the accuracy of the devices. During the first stage development and testing period, the sensors were validated in laboratory conditions using reference equipment that followed EU standards EN 14907:2005 (gravimetric device with filters that collects PM2.5). Later, during the second stage, the sensors were validated in real-life conditions. 

The network created started operating on 1 March 2019 and is still working 24/7 at the time of writing this manuscript. The real-time data for the city of Volos can be found at http://greenyourair.org/ (accessed on 30 October 2022).

### 2.3. Statistical Analysis

The independent samples t-test and analysis of variance were used for the between groups comparisons of continuous variables as appropriate. The chi-square test was used to identify possible relationships between categorical variables. We used regression analysis and Pearson’s correlation in order to describe the relationship between the number of ED visits and temperature, humidity, age and PM2.5 levels. The significance for all tests was set at *p* values < 0.05 and all tests were two-tailed. The SPSS statistical package was used for all statistical analyses (IBM Corp. Released 2016. IBM SPSS Statistics for Windows, Version 24.0. Armonk, NY, USA; IBM Corp).

## 3. Results

### 3.1. Patient Characteristics

During our study period, a total of 728 patients visited the adult ED for respiratory causes. The male to female ratio was 1.05 (373 male and 355 female patients), while the mean age for male patients was 64.50 ± 19.69, and for female patients, it was 61.43 ± 20.52.

A total of 310 patients (mean age (standard deviation): 66.35 ± 20.77; male gender: 166 (54%)) were diagnosed with an URI, 202 patients were diagnosed with a lower respiratory tract infection/pneumonia (mean age (standard deviation): 52.41 ± 20.70; male gender: 106(52%)), 53 patients were diagnosed with asthma exacerbation (mean age (standard deviation): 50.39 ± 20.05; male gender: 15(28%)) and 163 patients were diagnosed with COPD exacerbation (mean age (standard deviation): 69.49 ± 12.20; male gender: 85(52%)).

### 3.2. PM2.5 Levels

In the city of Volos, the mean annual PM2.5 concentration during the study period was calculated to be 30.03 ± 17.47 μg/m3 compared to WHO’s yearly limit of 10 μg/m3. Overall, the suggested daily limit of PM2.5 (25 μg/m^3^) was exceeded for 178 days, which was 48.60% of the study period. The recorded levels of PM2.5 were found to be higher in winter when compared with summer (mean difference: 25.64, *p*: < 0.001), autumn (mean difference: 20.20, *p*: < 0.001) and spring (mean difference: 17.71, *p*: < 0.001).

### 3.3. PM2.5 Levels, Humidity, Temperature, Age and ED Visits

The mean number of daily ED visits for respiratory causes was 1.99 ± 1.81. The number of monthly ED alongside the corresponding monthly PM2.5 levels are presented in Figure 2. As shown in Table 1, a 25.42% increase in daily ED visits for all respiratory causes was identified when PM2.5 levels exceeded the daily limit (1.77 vs. 2.22 visits per day; *p*: 0.018). Further analyses performed by season (Table 2) illustrated that the increase in ED visits was more pronounced during winter (34.08%) and autumn (29.40%), although the difference was non-statistically significant. Moreover, there was statistically significant elevation in ED visits when PM2.5 levels of the previous day were higher than the proposed limit 25 μg/m^3^ (*p* = 0.018).

Additionally, we identified that temperature was negatively correlated with ED visits (r: −0.21; *p* < 0.001), while humidity did not exhibit any statistically significant correlation (r: 0.03; *p* = 0.58). Finally, age was found to be positively correlated with ED visits (r: 0.69; *p* < 0.001).

### 3.4. Specific Conditions and PM2.5 Levels

A statistically significant increase in ED visits for URI was noted during the days that PM2.5 levels exceeded the limit of 25 μg/m^3^, when compared to the days when PM2.5 levels were below 25 μg/m^3^ (0.71 vs. 0.99 visits/day; *p* = 0.01). Table 1 presents the comparisons for all studied conditions. No statistically significant differences were identified when a further analysis was performed based on the season (Appendix A). Finally, no statistically significant differences in mean ED visits were observed for all studied conditions (URI: *p* = 0.05; Pneumonia: *p* = 0.42; Asthma: *p* = 0.28; COPD: *p* = 0.47) when comparing males and females.

In regression analysis, a linear correlation (r square: 0.022; *p* < 0.001) was noted between the levels of PM2.5 and the total number of daily ED visits. This is described by the following equation: total number of ED visits = 1.524 + 0.015 × PM2.5 levels (Figure 3). When we included temperature and humidity in the model’s parameters, the model was still statistically significant (r square: 0.047; *p* < 0.001) although the only parameter with statistical significance was temperature.

## 4. Discussion

In our study, PM2.5 levels exceeded the daily limit of 10 μg/m^3^ during 48.6% of the study period, with the mean PM2.5 concentration being 30.03 ± 17.47 μg/m^3^. PM2.5 levels were higher during the winter compared to autumn, spring, and summer. ED visits were significantly higher on days when PM2.5 concentrations exceeded the daily limit, or the day after. Although age and temperature had a significant correlation with ED visits, humidity did not play a role in the number of daily ED visits. Moreover, high PM2.5 levels were associated with an increase in URI-related ED visits.

Air pollution has become increasingly prevalent during the last decades, since many global areas are exposed daily to high levels of air pollutants [14]. While large metropolises suffer greatly [15], semi-urban and rural areas can also be affected [8]. Research pertaining to the relationship between air pollution and human health in rural and semi-urban areas may further highlight the importance of air pollutants on respiratory health.

Despite the fact that the city of Volos is a semi-urban area, high levels of air pollutants have been previously recorded [10,11]. In accordance with our study, Moustris et al. [10] reported elevated levels of PM10 pollution in the city of Volos, with PM10 levels regularly exceeding the daily and annual proposed limits by WHO. In the study by Moustris et al., an increase in the annual PM10 concentration in the city of Volos resulted in an increase in the annual hospital admissions for respiratory diseases. However, in our study, we researched the relationship between daily PM2.5 concentrations and ED visits for respiratory causes, which further highlights the direct effect of PM2.5 pollution and respiratory health.

PM2.5 production is mainly anthropogenic [14]. Fine particle matter is produced primarily through fuel and wood combustion, mainly deriving from vehicle and biofuel emissions [16]. A meta-analysis of PM source apportionment in Europe revealed that traffic, as well as wood burning during the cold months, were the most important factors in PM production [16]. In the last decade, the number of Greek households using wood burning as a means of heating during the winter rapidly increased due to the high petrol prices and economic crisis [17]. This increase in wood burning has subsequently led to an increase in fine PM levels, which are significantly higher during the cold months [17]. Likewise, the present study indicates that PM2.5 levels in the city of Volos are higher during winter compared to other seasons. This relationship may indicate that the production of PM2.5 by wood burning may be a significant component of environmental pollution in semi-urban or rural areas, where pollutant production by industrial processes is less prominent.

Approximately 4.2 premature million deaths annually can be attributed to PM2.5 pollution, placing fine particle pollution as the fifth most common cause of death worldwide [18]. Exposure to elevated PM2.5 levels has been associated with increased Out-of-Hospital Cardiac Arrests (OHCAs), as well as a variety of cardiovascular and respiratory diseases [19,20,21]. Interestingly, a meta-analysis by Atkinson et al. [20] demonstrated that PM2.5 pollution is more strongly associated with mortality due to respiratory causes [20]. Indeed, high PM2.5 levels are associated with increased rates of respiratory infections and asthma or COPD exacerbations, leading to an increase in ED visits for respiratory causes [22,23].

According to our findings, in the semi-urban city of Volos, there was a statistically significant association between PM2.5 levels and ED visits for respiratory causes. These findings are in accordance with two similar studies conducted in the city of Volos, which demonstrated that fine particulate matter pollution is associated with increased hospital admissions for respiratory causes in both adult and pediatric patients [10,11]. As previously mentioned, high levels of PM2.5 can precipitate a variety of respiratory diseases, such as respiratory infections and asthma and COPD exacerbations, as well as increased hospital admissions and mortality due to respiratory causes [20,22]. Specifically, our research team, using the model developed by the GreenYourAir research group, discovered that pediatric ED visits in the city of Volos were linearly correlated with PM2.5 levels [11]. In our study, the same research group discovered that a similar relationship exists in adult patients. Furthermore, our results revealed a significant correlation between age and ED visits for respiratory causes. As showcased in previous studies, the effect of PM2.5 on respiratory diseases seems to be more pronounced in children and adults [7,24] According to previous studies, the effect of PM2.5 on respiratory-related conditions is more significant in adults more than 65 years old [7].

Several meteorological factors have been shown to influence the concentration of PM2.5 and its effect on human health [25,26]. In our study, a decrease in the mean daily temperature resulted in an increase in daily ED visits while humidity did not have a statistically significant correlation with ED visits. Similarly, a study by Wang et al. in 28 Chinese cities showcased that low temperatures during winter were correlated with higher PM2.5 concentrations [25]. The researchers hypothesized that this effect may be due to the increased coal and wood burning during the cold months. Moreover, low temperature has been associated with an increased susceptibility to URIs [27], which may increase respiratory-related ED visits. However, a study in Lima, Peru, showcased that the effect of PM2.5 concentrations in respiratory and cardiac mortality was more pronounced when the mean daily temperature was higher than 23.8 °C [28]. A similar relationship between PM10 and respiratory disease was found in a systematic review and meta-analysis by Chen et al. [29], which found that the effect of air pollution on respiratory diseases is more significant when the temperature is higher. These conflicting results prove that more research is necessary in order to accurately understand the complicated effects of meteorological factors in fine particulate matter concentrations and their effects on human health. Moreover, while our results did not reveal a significant correlation between humidity and ED visits, it is important to note that several researchers proved that high relative humidity may be accompanied by higher PM2.5 and PM10 concentrations, which may have a negative effect on respiratory-related ED visits [30,31].

In our study, there was a statistically significant association between URI-related ED visits and PM2.5 levels. The association between PM2.5 levels and respiratory infections has been proved by numerous studies, which showcase that increased PM2.5 levels lead to increased ED visits and hospital admissions for upper and lower respiratory infections [7,32]. The study by our research team in the city of Volos, also indicated that high daily PM2.5 levels are correlated with increased pediatric ED visits for URI [11]. This effect has been attributed to increased susceptibility to infections [33] based on experimental animal models [34,35]. While the exact mechanism remains unknown, it is speculated that PM2.5 impairs the host’s defense of the respiratory system by altering epithelial cell functions and immune cell activity [33]. This effect is more pronounced in pediatric and elderly patients [33].

Τhe association between PM2.5 pollution and chronic respiratory diseases has been well documented. Both PM2.5 and PM10 may cause COPD exacerbations, leading to more ED visits for respiratory causes and increased morbidity and mortality in COPD patients [22,36]. Interestingly, it has been demonstrated that PM2.5 exposure may cause chronic respiratory dysfunction, creating emphysematous lesions and chronic inflammation, which in turn may lead to COPD developments [36]. Moreover, PM2.5 pollution can aggravate the effects of smoking on lung function, increasing the likelihood of COPD [36]. In contrast to the existing literature, no statistically significant association between PM2.5 levels and COPD association was noted in our study. However, this discrepancy may be attributed to the small number of patients presenting to the ED with respiratory symptoms due to COPD exacerbation.

PM2.5 pollution may also worsen asthma symptoms, leading to increased ED visits and hospital admissions for asthma exacerbations [4,37,38]. A meta-analysis by Fan et al. demonstrated that asthma-related ED visits increased proportionally to PM2.5 levels [4]. The effect was more pronounced in pediatric patients. Moreover, asthma exacerbations are most likely to occur during spring possibly due to the prevalence of allergens, such as pollen [37]. The association between PM2.5 pollution and asthma can be attributed to the increased inflammation of airway epithelial cells and the increased secretion of inflammatory cytokines [39,40]. Our study did not reveal a statistically significant association between PM2.5 levels and asthma-related ED visits. Moreover, our team in a similar study in pediatric patients did not discover a statistically significant relationship between asthma-related ED visits and PM2.5 levels. Similarly to COPD cases, this could be explained by the small number of asthma-related ED visits during our study period.

### Limitations

Our study has some limitations that should be acknowledged. Firstly, is it a single-center study that collected data from a single ED; thus, our results should be interpreted with caution. Moreover, the diagnoses were made in an emergency setting; therefore, there may have been some misdiagnoses. It is also important to note that our analysis was conducted based on the mean PM2.5 levels of each day and may not accurately reflect patient exposure, since no regional analysis was conducted.

## 5. Conclusions

In our study, high PM2.5 levels were associated with an increase in adult ED visits for respiratory causes. PM2.5 pollution was statistically significantly related to the number of URI-related visits but not COPD or asthma-related visits. Moreover, low temperatures and increased age increased ED visits for respiratory causes. While the association between fine particle pollution and respiratory illnesses has been well documented, more studies are necessary in order to ascertain the pathophysiological mechanisms leading to respiratory dysfunction, as well as individual factors that may predispose certain patients to PM2.5-related respiratory symptoms.

## Figures and Tables

**Figure 1 jpm-12-01849-f001:**
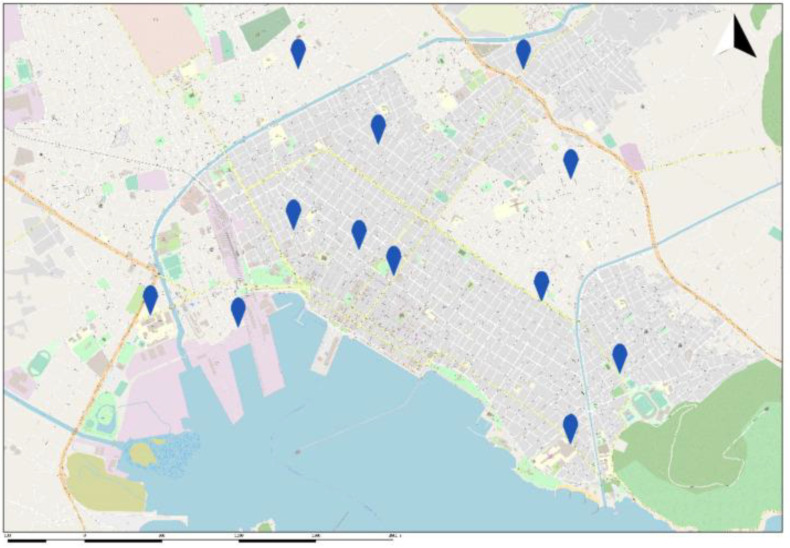
Map of the city of Volos presenting the location of sensors. Markers indicate the location of the sensors.

**Figure 2 jpm-12-01849-f002:**
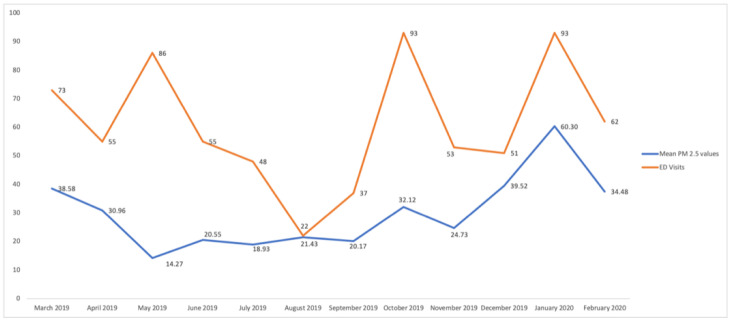
Monthly PM2.5 levels and ED visits.

**Figure 3 jpm-12-01849-f003:**
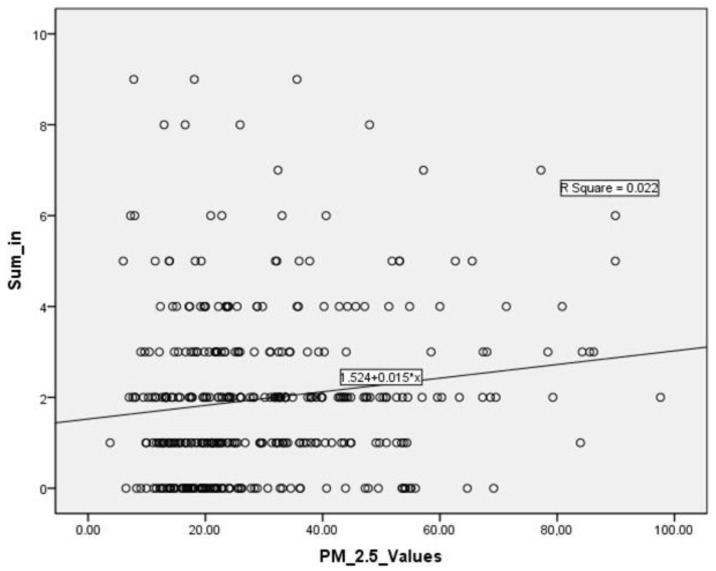
Correlation between daily PM2.5 levels and daily ED visits.

**Table 1 jpm-12-01849-t001:** Table presenting the number of ED visits in relation to daily PM2.5 levels.

	PM2.5 Level	Total Days N = 366	Number of ED Visits N = 728	Mean ED Visits/Day ± SD	% Increase in Mean ED Visits	*p* Value
All patients	<25 μg/m^3^	188	333	1.77 ± 1.81	25.42%	0.018
	≥25 μg/m^3^	178	395	2.22 ± 1.77		
URI	<25 μg/m^3^	188	134	0.71 ± 1.03	38.72%	0.011
	≥25 μg/m^3^	178	176	0.99 ± 1.03		
Pneumonia	<25 μg/m^3^	188	93	0.49 ± 0.91	23.79%	0.224
	≥25 μg/m^3^	178	109	0.61 ± 0.94		
Asthma exacerbation	<25 μg/m^3^	188	30	0.16 ± 0.43	−19.03%	0.465
	≥25 μg/m^3^	178	23	0.13 ± 0.35		
COPD exacerbation	<25 μg/m^3^	188	76	0.40 ± 0.68	20.90%	0.261
	≥25 μg/m^3^	178	87	0.49 ± 0.75		

Abbreviations: ED: emergency department; SD: standard deviation; URI: upper respiratory tract infection; COPD: Chronic Obstructive Pulmonary Disease.

**Table 2 jpm-12-01849-t002:** Table presenting the number of ED visits in relation to mean PM2.5 levels of each season.

	PM2.5 Level	Total Days	Number of ED Visits	Mean ED Visits/Day ± SD	% Increase in Mean ED Visits	*p* Value
Winter	<25 μg/m^3^	18	32	1.78 ± 1.48	34.08%	0.12
	≥25 μg/m^3^	73	174	2.38 ± 1.77		
Spring	<25 μg/m^3^	42	102	2.43 ± 2.32	−7.76%	0.65
	≥25 μg/m^3^	50	112	2.24 ± 1.67		
Summer	<25 μg/m^3^	74	102	1.38 ± 1.41	−7.30%	0.78
	≥25 μg/m^3^	18	23	1.28 ± 1.13		
Autumn	<25 μg/m^3^	54	97	1.80 ± 1.86	29.40%	0.21

Abbreviations: ED: emergency department; SD: standard deviation.

## Data Availability

The data that support the findings of this study are available from the corresponding author (MM) upon reasonable request.

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
