# Peer review of "Effect of PM2.5 Levels on ED Visits for Respiratory Causes in a Greek Semi-Urban Area"

_jpm, 2022, doi:10.3390/jpm12111849_

Round 1
Reviewer 1 Report
This paper use the 24-hour PM2.5 air pollution data were prospectively collected 19 from twelve fully automated air quality monitoring stations to investigate the correlation between PM2.5 levels in 16 the semi-urban city of Volos and Emergency Department (ED) visits for respiratory causes.
I think there are several serious problems in this paper:
1. The first paragraph is to discuss the principle of PM2.5 causing respiratory diseases, and does not give the situation of respiratory diseases in Volos. This leaves a poor understanding of the extent of respiratory disease in the city.
2. The second paragraph of the preface simply lists the relevant studies on the relationship between PM2.5 and respiratory diseases. The design, methodological flaws and gaps of such studies were not pointed out.
3. Please give the necessary references in the first paragraph of the Materials and methods section. No inclusion criteria were given for the study subjects.
4. Why not consider other pollutants; please also give site distribution; did the collected population live near the site? If not, how to ensure that the concentration of pollutants near the individual is consistent with the site?
5. The results of the study lack sensitivity analysis results. No correction for the effects of other contaminants.
6. The discussion section is not sufficient to clarify the gap between this study and previous studies and the medical value.
Author Response
This paper use the 24-hour PM2.5 air pollution data were prospectively collected 19 from twelve fully automated air quality monitoring stations to investigate the correlation between PM2.5 levels in 16 the semi-urban city of Volos and Emergency Department (ED) visits for respiratory causes.
I think there are several serious problems in this paper:
Thank you very much for devoting your time to reviewing our manuscript. We hope this revision will satisfy your high standards and will consider to accept this paper for publication.
- The first paragraph is to discuss the principle of PM2.5 causing respiratory diseases, and does not give the situation of respiratory diseases in Volos. This leaves a poor understanding of the extent of respiratory disease in the city.
Thank you very much for this comment. We have provided more information concerning the extent of respiratory disease in the city of Volos.
- The second paragraph of the preface simply lists the relevant studies on the relationship between PM2.5 and respiratory diseases. The design, methodological flaws and gaps of such studies were not pointed out.
Thank you for your comment. We have provided more information about the studies presented in the introduction section of the manuscript.
- Please give the necessary references in the first paragraph of the Materials and methods section. No inclusion criteria were given for the study subjects.
Thank you for this comment. We have added the necessary references in the Materials and methods section, as well as the inclusion criteria for the study subjects.
- Why not consider other pollutants; please also give site distribution; did the collected population live near the site? If not, how to ensure that the concentration of pollutants near the individual is consistent with the site?
Thank you very much for your comment. We have added more information regarding site contribution in the Materials and Methods section of our manuscript.
- The results of the study lack sensitivity analysis results. No correction for the effects of other contaminants.
Thank you for this comment. We have added sensitivity analysis about . Unfortunately, our study was designed to account only for PM2.5 concentrations in the city of Volos and there is no available data to account for other pollutants.
- The discussion section is not sufficient to clarify the gap between this study and previous studies and the medical value.
Thank you very much for this comment. We have altered our Discussion section, trying to highlight the importance of this study and its differences with other available literature.
Reviewer 2 Report
This manuscript reported the effects of PM2.5 exposure on ED visits of respiratory diseases in the city of Volos. The ED visits for various diseases were compared between PM2.5>= 25μg/m3 and < 25μg/m3. And the results suggested that PM2.5 levels were significantly associated with ED visits for respiratory causes. I have a few suggestions for the authors to consider. 1. Both PM2.5 levels ED visits are continuous variables, the association between PM2.5 levels and ED visits could be directly evaluated using continuous analysis model such as linear mixed model. 2. Ambient temperature is an important factor affecting ED visits for respiratory diseases, and this study also found that the number of ED visits was higher in winter when compared with those in spring or summer. It is necessary to further analyze whether there is an interaction between ambient temperature and PM2.5 on ED visits. 3. Whether there are age or gender differences modify the association of PM2.5 with ED visits? Here need more analysis.
Author Response
This manuscript reported the effects of PM2.5 exposure on ED visits of respiratory diseases in the city of Volos. The ED visits for various diseases were compared between PM2.5>= 25μg/m3 and < 25μg/m3. And the results suggested that PM2.5 levels were significantly associated with ED visits for respiratory causes. I have a few suggestions for the authors to consider.
Thank you very much for the time and effort you took to review our manuscript. We have revised our paper according to your comments. We hope that this revised version will satisfy your high standards you will consider to accept our article for publication.
- Both PM2.5 levels ED visits are continuous variables, the association between PM2.5 levels and ED visits could be directly evaluated using continuous analysis model such as linear mixed model.
Thank you for your suggestion. We elected to perform a linear regression analysis, as we feel that the model created better represents our data. We are already collecting more data to validate the identified relationship between PM2.5 and ED visits.
- Ambient temperature is an important factor affecting ED visits for respiratory diseases, and this study also found that the number of ED visits was higher in winter when compared with those in spring or summer. It is necessary to further analyze whether there is an interaction between ambient temperature and PM2.5 on ED visits.
Thank you very much for your comment. We have added the requested analysis about temperature, as well as a section about humidity.
- Whether there are age or gender differences modify the association of PM2.5 with ED visits? Here need more analysis.
Thank you for this comment. We have added an additional analysis concerning the effect of age on ED visits.
Round 2
Reviewer 1 Report
The article still considers only the relationship between PM2.5 and ED visits. The concentration of other pollutants may also affect ED visits. How to prove that ED visits's growth is only related to PM2.5, it is not reflected in the article.
Reviewer 2 Report
The authors has improved their work and all my previous questions have been answered.